# Development and Validation of an Examination Protocol for Arthroscopic Evaluation of the Temporomandibular Joint in Dogs

**DOI:** 10.3390/ani14091338

**Published:** 2024-04-29

**Authors:** Ina Quadflieg, Jasmin Ordobazari, Matthias Lüpke, Fritjof Freise, Holger A. Volk, Benjamin Metje

**Affiliations:** 1Department of Small Animal Medicine and Surgery, University of Veterinary Medicine Hannover, 30559 Hannover, Germany; ina.quadflieg@tiho-hannover.de (I.Q.); jasmin.ordobazari@tiho-hannover.de (J.O.); holger.volk@tiho-hannover.de (H.A.V.); 2Department of General Radiology and Medical Physics, University of Veterinary Medicine Hannover, 30173 Hannover, Germany; matthias.luepke@tiho-hannover.de; 3Department of Biometry, Epidemiology and Information Processing, University of Veterinary Medicine Hannover, 30559 Hannover, Germany; fritjof.freise@tiho-hannover.de

**Keywords:** dog, veterinary surgery, arthroscopy, temporomandibular joint, needle arthroscopy

## Abstract

**Simple Summary:**

Temporomandibular joint (TMJ) disorders are often undiagnosed in small animal practice. One reason is that there is a sparsity of targeted diagnostics and many fail to categorise any meniscal changes. Temporomandibular joint arthroscopy offers a unique opportunity to gain a direct insight into the joint and its structures. Arthroscopy offers the possibility to visualise and identify pathological conditions and, if necessary, surgical intervention and sampling of the joint can be carried out directly. However, the authors are not aware of an established standardised examination scheme for temporomandibular joint arthroscopy in dogs. Therefore, a standardised examination scheme was developed and tested in this study. The examination scheme was advantageous as it allowed a thorough assessment of most structures, which should have been visualised. The established guidelines can provide clinicians with a standardised arthroscopy scheme for the temporomandibular joint.

**Abstract:**

Due to the previously limited intra-articular diagnostic possibilities of the canine temporomandibular joint, an examination protocol for the canine temporomandibular joint (TMJ) was developed and tested in this study using a needle arthroscope. In total, the discotemporal (DTJ) and discomandibular (DMJ) joint compartments of 32 animals (64 TMJs) were examined arthroscopically. During the examinations, 15 anatomical landmarks per joint side were evaluated in regard to their visibility and accessibility. All arthroscopies were performed by the same examiner and the same assistant to ensure standard methods were applied. The examination procedure which was developed here proved to be a reliable tool for examining this joint. The 15 anatomical landmarks that were to be examined could be reliably visualised and assessed in all TMJs with a certainty of 86% to 100% by both observers. This tool provides clinicians with a reliable examination aid for everyday practice and ensures the comparability of results. In the future, this could provide an opportunity to better diagnose and treat TMJ pathologies.

## 1. Introduction

Temporomandibular joint (TMJ) disorders can cause considerable pain and have a massive impact on the quality of life of the affected animal [1,2]. As in humans, TMJ disorders in dogs are still a presumably underdiagnosed area of medicine and available studies on this topic are scarce [3,4,5]. Often, various TMJ disorders in canine patients occur in combination and require medical or surgical treatment [3].

The TMJ is composed of the junction between the *Os temporale* and the *Ramus mandibulae*. It is a synovial joint consisting of the larger, dorsal discotemporal joint (DTJ) and the smaller, ventral discomandibular joint (DMJ) [6,7,8,9]. The *Discus articularis* divides the TMJ into two separate joint cavities. A thin *Capsula articularis*, the *Ligamentum laterale*, and the surrounding soft tissue serve to stabilise the joint [6,10,11,12]. The precise control of jaw movements, necessary for predators to eat, play, and work, and necessary for communication in dogs, is maintained through the interaction of muscles, tendons, and ligaments [8]. The health and functionality of the TMJ therefore significantly influence well-being and quality of life [1,2].

However, a wide variety of pathologies can affect the TMJ. Traumatic injuries, degenerative joint disease, inflammatory processes, or congenital anomalies can lead to pain, restricted movement, and dysfunction [11]. Diagnosing these conditions can be challenging for the examiner, as the TMJ is often difficult to access using conventional diagnostic techniques due to its complexity and localisation [13].

Therefore, diagnostic imaging such as magnetic resonance imaging (MRI) and computed tomography (CT) is essential in the evaluation of TMJ disorders [3,10,13]. The use of CT diagnostics, and particularly cone beam computed tomography (CBCT), has become established in TMJ pathology. This diagnostic tool enables the precise assessment of bony structures, allowing for the diagnosis of fractures, bone formations, narrowing of the joint space, and chondral and subchondral bone lesions [9,14]. One disadvantage of CT diagnostics is its limited ability to assess soft tissue. This diagnostic tool cannot adequately assess the surrounding tissue, particularly the intervertebral disc due to its narrow texture, as well as the joint cavity and synovium [10].

To adequately assess these structures, further diagnostic options are necessary. In 1970, TMJ puncture and arthroscopy were first documented as diagnostic tools for humans [15,16]. These minimally invasive methods have proven to be effective. The procedures remain highly useful today, as they allow for direct visualisation of the joint and assessment of the soft tissue, particularly the articular disc. Additionally, the cartilage surface structures of the joint and the synovium can be examined. Pathological conditions can be detected directly, allowing for an accurate diagnosis. If necessary, surgical intervention can be carried out in the same treatment step [17,18]. Although TMJ arthroscopy has been performed in experimental studies in dogs, there is no description of the anatomical structures that can be visualised and assessed in dogs [19,20]. Similarly, the authors are not aware of an established standardised examination scheme as described for horses [21]. The anatomical structures that can be visualised and assessed in dogs have not yet been described.

For this reason, this study includes the comparative, arthroscopic assessment of the TMJ joint compartments of dogs. The aim was to develop and test for the first time a standardised examination scheme for arthroscopy of the TMJ in dogs. This should ensure a uniform and reliable assessment of this examination method. The examiner is thus provided with an examination aid that can be used safely in everyday clinical practice and ensures comparability of the examination results. In the future, this could provide an option to better diagnose and treat TMJ pathologies.

## 2. Materials and Methods

The study population included 32 cadaver heads from 14 females (10 spayed, 4 intact) and 18 males (5 spayed, 13 intact), with an age range of 2–15 (median 8.25) years. The animals were euthanised without any relation to this study. Subsequently, the owners consented and released the bodies for scientific purposes at the Department of Small Animal Medicine and Surgery of the University of Veterinary Medicine Hannover. The study was conducted under the university law of Lower Saxony (Germany) approved by the animal welfare officers of the dissertation committee. All animals had a minimum weight of 20 kg (20.1 kg–59.2 kg, median: 32.55 kg). The following were included: Bearded Collie (n = 1), Bernese Mountain Dog (n = 1), Bordeaux mastiff (n = 1), Border Collies (n = 2), Boxers (n = 2), Double Doodle (n = 1), Elo (n = 1), Flat Coated Retriever (n = 1), Galgo Español (n = 1), Golden Retrievers (n = 2), Irish Setter (n = 1), Labradoodle (n = 1), Labrador Retrievers (n = 5), Longhaired Collie (n = 1), Crossbreeds (n = 4), Rhodesian Ridgebacks (n = 4), German Shepherds (n = 2), Shar-pei (n = 1).

The study includes the arthroscopic assessment of the canine TMJ. For this purpose, the DTJ compartment and the DMJ compartment of the two TMJs were examined arthroscopically. The examinations were carried out using a needle arthroscope (Arthrex, Naples, FL, USA, NanoScope Handpiece (AR-3210-0044) and sleeve (AR-3210-0054)) with a handpiece diameter of 1.9 mm and an outer diameter of 2.4 mm for the sleeve, as well as 0° optics.

To make the results of the study objectively comparable, the animals were examined arthroscopically in batches. Beforehand, they were stored frozen at −19 °C and thawed at room temperature for approx. 40–48 h before the assessment to ensure similar conditions. First, the animals were prepared for surgery by shaving large areas around the TMJ. Afterwards, the animals were placed on the operating table in a lateral position and the head and neck region were lifted slightly to the side. To ensure an optimal view during the examination of this joint, the jaws of the animals were distracted. For this purpose, a constant force of 50–55 newtons was applied to the TMJ during the procedure. The technique was used to improve the quality of the examination. The relatively low force setting is based on previous studies that show significantly higher force measurements in the mouth and jaw area during daily use of these structures [22,23,24]. A special instrument was used for this purpose, consisting of a quick-grip clamp (Connex, COX864630, Celle, Germany) and custom-made splints for the upper and lower jaw (3D printer: Kühling&Kühling (HT500.2, Trappenkamp, Germany), printing material: Extrudr, (GreenTEC Pro, Lauterach, Austria)). The maxillary splint was placed on the nasal bone of the maxilla and the mandibular splint was placed at the level of the mandibular canines (Figure 1).

A force sensor (Variohm Eurosensor, Towcester, UK, FX1901) was installed in the upper jaw mould for permanent measurement of the force exerted on the TMJ. This force sensor was connected to a measurement transmission device (National Instruments, Austin, TX, USA, cDAQ-9171, 195724D-01L and integrated NI, 9219 Universal Analogue Input Module, Series C, four channels). The force measurements were visualised with the help of the measuring program Dasylab (Version 2020) on a laptop computer (Lenovo ideapad320, Beijing, China) (Figure 2).

To ensure a uniform force on the TMJ, the law of leverage (F_1_ × a_1_ = F_2_ × a_2_) was applied for this purpose. The corresponding anatomical structures were measured in each animal as follows. First, a tubular element with a diameter of 1 cm was utilised, which was modified with insulating tape (Tesa, Norderstedt, Germany, Iso Tape, 10 m × 15 mm) for better slip resistance in the oral cavity. This tubular element was then fixed between the first molar tooth of the lower jaw and the fourth premolar tooth of the upper jaw on the side of the TMJ joint to be examined. The distance (‘force arm’ = a_1_) from the force applied to the jaw section to the pivot point (tubular element) was measured and titled a_1_. Furthermore, the distance (‘load arm’ = a_2_) from the pivot point to the TMJ was measured and named a_2_ (Figure 3). Using these values, the law of leverage was used to calculate the corresponding force that had to be applied to the animal’s TMJ to ensure a uniform force of 50–55 newtons for each examination. The quick-grip clamp with the integrated force sensor made it possible to apply the correspondingly calculated force constantly and adjust it if necessary.

A standard surgical approach was then performed from the lateral side as described in previous studies [10,25]. This step of the examination was conducted to initially facilitate the locating and penetration of the respective temporomandibular joint cavity. However, the subsequent described joint access was minimised to allow solely for the penetration of the arthroscope into the joint. First, a scalpel blade (No. 15) was used to make a skin incision caudoventral to the *Arcus zygomaticus*. The *Musculus masseter* was exposed by bluntly dissecting the overlying muscles. Subsequently, the *Musculus masseter* was also elevated at its ventral point of attachment to the *Arcus zygomaticus* by blunt dissection. The buccal and auriculopalpebral nerve sections, along with the *Glandula parotis* and its duct *(Ductus paroticus*), were omitted and retracted. After visualising the *Ligamentum laterale*, the TMJ, and the *Capsula articularis* of the joint, a horizontal incision was made through the aforementioned structures. The *Discus articularis* of the TMJ was detached at its lateral attachment of the superior as well as inferior articular cavity to allow complete access to these compartments (Figure 4).

Bilaterally, both TMJ cavities of each specimen were arthroscopically examined in a randomised order and subsequently compared. For descriptive comparability of the examination, 15 anatomical structures were identified for each TMJ, which were examined for their visibility and accessibility during the arthroscopy. The former described equine TMJ arthroscopy scheme was used as a blueprint [21]. In each case, nine different structures were evaluated in the upper compartment and six anatomical landmarks were analysed in the lower compartment. In total, 960 anatomical landmarks were reviewed in this study (Table 1).

Data were recorded in a tabulate fashion using the standardised protocol developed for this purpose as visible or not visible. Images were taken for documentation.

During the examination, the dog’s TMJ was irrigated with a 0.9% NaCl solution and the irrigation attachment (2.4 mm ⌀) of the needle arthroscope to improve visibility. The TMJs were viewed in a randomised order, with either the left or right joint examined first.

During the examination procedure, the investigators proceeded as follows: First, the arthroscope was inserted into the DTJ through the lateral approach and the centre of the joint was examined first, examining the *Fossa mandibularis*, articular cartilage, and the *Discus articularis*. This was followed by an inspection of the most rostral and caudal aspect of the joint, examining the attachment of the *Discus articularis* at both landmarks. The arthroscope was now inserted further medially into the joint at the midpoint, giving the investigators an overview of the medial attachment of the discus, the retro-discal soft tissue, and by advancing rostrally past the medial attachment of the discus, an overview of the medial aspect of the joint. If an anatomical structure to be assessed could not be visualised right away, the examination was continued according to the above scheme and then the structure still to be assessed was visited again (Figure 5).

After the comprehensive inspection of the DTJ compartment, the DMJ compartment was examined, whereby the arthroscope was first reintroduced centrally into the joint via the lateral approach. The *Discus articularis* and the *Fossa mandibularis* were examined. This was followed by an assessment of the rostral aspect of the joint, the caudal synovial folds, and finally the medial aspect of the joint with the retro-discal soft tissue (Figure 6).

All arthroscopies were performed by the same examiner (observer 1) and assistant (observer 2) to ensure consistency of the results. Both examiners had a similar wealth of experience in performing arthroscopies and first had to learn the procedure for this examination method. During the examination, both examiners independently recorded the visibility of anatomical structures in tabular form, distant from each other. There was no communication between the examiners during the individual examinations to avoid any influence on the results by the individual investigator. After each examination, it was determined whether the examiners could assess the selected structures, in order to potentially increase the force applied to the jaw if necessary. If the structure under examination could not be adequately assessed by both observers with a previously described force application of 50–55 newtons on the TMJ, the second step was to increase the force application to 100 newtons in this joint to obtain a possibly better view with this higher-force application.

### Statistics Analysis

After the investigations, the data were entered into a suitable program (Microsoft Excel, Version 2309, 2023) and then analysed statistically (SAS, Enterprise Guide 7.15 and SAS software, version 9.4). First, a descriptive evaluation of the data was carried out to ensure a percentage representation of the apparent structures. This was followed by the Fisher test, which was used to check for differences in the lateral comparison between the right and left sides of the TMJ. In addition, a Cohen’s kappa was calculated along with a 95% confidence interval to determine agreement between the two observers. A *p*-value of <0.05 was considered significant.

## 3. Results

A total of 64 TMJs from 32 dogs were examined in this study. The DTJ and DMJ of each side of the TMJ were examined and checked for the visibility of the selected anatomical structures. In the DTJ, the following structures were assessed: rostral aspect of the joint, medial aspect of the joint, caudal aspect of the joint, *Fossa mandibularis* of the *Os temporale*, *Discus articularis*, retro-discal tissue, rostral attachment of the *Discus articularis*, medial attachment of the *Discus articularis*, caudal attachment of the *Discus articularis*. Furthermore, in the DMJ, the following structures were investigated: rostral aspect of the joint, medial aspect of the joint, caudal synovial plicae, caput mandibulae, *Discus articularis*, retro-discal tissue. In total, 960 anatomical structures were examined during this experimental procedure. As can be seen in Table 1, seven anatomical structures could be visualised in all dogs examined (100%). The remaining eight anatomical landmarks were also well identified (86–98%). The most difficult aspect of the examination was the visualization of the medial aspect of the joint, with its being identified at 92% by observer 1 and at 86% by observer 2.

In summary, observer 1 was able to evaluate all 15 anatomical landmarks in 25 dogs. Observer 2 was also able to see and record all landmarks in 21 canine heads. For the remaining animals, a maximum of five landmarks per animal were not identified.

It was noticeable that the examinations were perceived to be more difficult in animals with very pronounced muscle structures in the area of the TMJ. First of all, it was difficult for the investigators to expose the TMJ and, in addition, the joint space of these animals was very narrow. Therefore, accessing and entering the joint with the arthroscope was a challenge for the examiner. Due to the limited accessibility, it was most difficult to visualise the medial aspect of the two joints, as this required a deeper access into the joint.

In addition, in several of the animals examined, abnormalities of the articular cartilage in the TMJ cavities were observed, which was mainly evident in the dissolution and detachment of the articular cartilage of the joint (Figure 7).

If an anatomical structure could not be sufficiently identified by both observers, the force was adjusted to 100 newtons, so that it could be evaluated whether an increase in force produced the desired success. Overall, by increasing the force in three animals, the visibility of structures for observer 1 was increased from 939 structures seen to 945 structures and for observer 2 from 934 to 940 structures seen. With the assistance of this force increase, n = 6 more structures were accessible to both observers (Table 2).

The visibility of the anatomical landmarks did not differ between the left and right TMJ sides (*p*-values ranging from 0.1191 to 1). The interobserver comparison, which was statistically examined using Cohen’s kappa, also showed a high agreement of the results (0.8474, 95% confidence interval: [0.7360, 0.9588]). Overall, the observers disagreed only in seven of the 960 cases. Due to this small number, more detailed calculations of Cohen’s kappa, for example grouped by structure, are not reported.

## 4. Discussion

The objective was to create a standardised assessment procedure for the examination of the TMJ and to describe a standardised guide for its arthroscopy, as has been formerly described for the horse [21]. With the assessment procedure of the TMJ in the current study, the majority of the anatomical structures were visualised. Seven of the 15 structures could be visualised completely while the remaining eight landmarks could be assessed more than 85% of the time. The medial aspect of the DTJ and the DTM presented a particular challenge for the investigators. In total, the two observers visualised 59 and 55 structures in the DTJ, respectively, and 60 landmarks in the DTM, during 64 examinations.

Osteoarthritis is the most common disease of the TMJ in dogs. The pathogenesis of TMJ osteoarthritis is characterised by erosion, degenerative changes and abrasion of the fibrocartilage, as well as local thickening and remodelling of the subchondral bone with the development of osteophytic outgrowths [3,26].

The medial side of the temporomandibular joint is more frequently affected than the lateral side, with more osteophytes forming in the medial region and cystic structures also being observed. Animals with this disease often show a biaxial narrowing of the joint space, with this narrowing occurring mainly on the medial side if the disease is unilateral.

Therefore, it is crucial to objectively assess the medial aspect of the temporomandibular joint to evaluate the mentioned pathological processes. However, examining this area can be challenging for the examiner due to its location and accessibility. With our proposed examination technique, we were able to visualise this area in a minimum of 86% of the cases.

No significant difference was found between diseased structures of the TMJ on the side of the DTJ and on the side of the DMJ. Therefore, the evaluation of both TMJ compartments is of equal importance. TMJ fractures after trauma occur equally frequently in both the mandibular fossa and the mandibular caput, making appropriate assessment of both bony structures essential [3]. During CT diagnostics, the entire cranial bone can be externally visualised in a 3D representation, whereas arthroscopy offers the crucial advantage of direct visual assessment within the joint. The arthroscopy examination allows for a direct assessment of the integrity of the joint surface, articular cartilage, which covers the bone, soft tissue, and particularly the articular disc. Furthermore, treatment measures can be carried out directly and possible bone fragments can be removed. However, it should be noted that, in contrast to CT diagnostics, arthroscopy is an invasive examination method [10,17].

The needle arthroscope used for this purpose, with a diameter of 1.9 mm (NonoScope sleeve (diameter 2.4 mm)) and an optic of 0°, proved to be a reliable examination instrument. The small diameter of the arthroscope provided the advantage of being able to adequately examine even narrow joint compartments. Since the interobserver comparison also showed remarkable agreement, it can be assumed that this examination scheme is easily transferable and thus individually applicable.

In the current study, the left and right TMJ were equally accessible, and no significant side difference was detected in the assessment of the anatomical structures. Prior randomisation of the order of the joints to be examined also reduced a potential bias in test conditions.

To ensure optimal visibility during the experiment, a consistent force ranging from 50 to 55 newtons was applied to the TMJ as described above. Previous studies suggest that the forces exerted on the mouth and jaw area in dogs during everyday activities exceed 100 newtons by a significant margin. Thus, it can be inferred that the force applied for TMJ evaluation should not pose any harm to the patient [22,23,24,27]. Nevertheless, it must be mentioned that the application of force through the use of the lever set, and the resulting distraction of the joint, does not represent a direct physiological application of force and can therefore result in an altered force effect on the TMJ and the remaining oral cavity structures.

No objective evidence of possible pathological effects on the cadaver heads during the studies was found. A subject for further investigation could be whether comparable results can be obtained without distracting this joint, as this would considerably simplify the procedure and make it easier to implement into everyday practice. In the present study, this force application was used to initially ensure simplified and standardised access to the TMJ.

An increase in force to 100 newtons was applied in case structures that needed to be assessed and could not be seen by both observers. However, the study found that only six additional landmarks could be detected with the increased force. It should be noted that the initially selected lower force of 50–55 newtons was sufficient to display the desired anatomical structures in most studies.

In the beginning, it became apparent that due to the complex structure of the TMJ and the small size of the structures to be examined, access to the selected joint compartment and visualisation of the existing anatomical structures was sometimes difficult. Thanks to the rapidly increasing routine in this diagnostic procedure, the respective jaw compartment could be precisely selected and examined after a brief time. The optics of the selected needle arthroscope provided the examiners with a clear view to effectively identify and assess the TMJ structures. The 0° optics and easy handling of the device enabled even the initially less experienced examiners to quickly orientate themselves in the joint.

The arthroscopic examination revealed a peculiarity, especially in dogs with a strong muscular expression in the region covering the TMJ. It was found that access to these joints was difficult due to the sizable muscular components around this joint. In these animals, the joint cavities appeared to be remarkably narrow, making arthroscopy of deeper or already narrow structures such as the medial aspect of the joint more difficult. The distraction of the jaw, which was always performed before the examination to improve the view of the joint, may also have been made more difficult by the pronounced musculature. This could have further impaired the visibility of the structures during TMJ arthroscopy.

Although it was noticeable that all four Rhodesian Ridgebacks proved difficult for the examiner to approach the joint, this is probably due to the well-preserved physical condition of these animals, which without exception showed a distinctly good musculature and should be interpreted less as a breed disposition. Since only four Rhodesian Ridgebacks were examined in this study, it is recommended that a further investigation be carried out with a higher number of animals and in particular specific breeds to investigate this hypothesis.

In addition, some studies have shown that there can be different characteristics of the TMJ within the skull morphologies and skull sizes. However, it has been found that small brachycephalic dogs are particularly affected and can suffer from incongruent temporomandibular joints and a poorly developed or absent retroarticular process. This can lead to laxity of the temporomandibular joint, as the stability of the temporomandibular joint is reduced [5,28]. It can be assumed that this facilitates access to the joint, as the existing TMJ structures are more easily accessible based on this reshaping and increased laxity. Therefore, a follow-up arthroscopic study with small dogs would be useful to investigate this hypothesis. The dogs included in this study all had well-integrated temporomandibular joints, so arthroscopic examination of these animals was probably not affected or influenced by breed.

The needle arthroscope offers clear advantages over the standard arthroscope due to its significantly smaller diameter, greater flexibility and mobility, and easier examination options due to the portable equipment [17]. However, the results of comparative studies on these two arthroscopic examination procedures are inconsistent. While some studies in dogs and horses show that the needle arthroscope facilitates detection and visibility, especially in narrow joints and small structures, other studies report poorer results with the needle arthroscope compared to the standard arthroscope due to the apparently higher sensitivity and manoeuvrability of the device and the reduced image quality described above [17,21,29,30,31,32].

In the authors’ assessment, the image quality in this study was sufficiently good to allow a detailed analysis of the examined areas. However, the handling of the needle arthroscope requires exceptional care, as it may be subject to rapid deformation due to its high flexibility. The needle arthroscope has demonstrated no adverse effects with careful handling. On the contrary, the device’s sensitive properties provide significant benefits to the operator.

The optics of an arthroscope also play an important role in the successful examination. For this study, a 0° optic was used, which allows for quick and simplified orientation in the joint. However, a previous comparative study in horses reported that the lateral joint compartments in particular may benefit from a 30° optic, as this facilitates viewing of the lateral structures [21]. The authors are convinced that a 30° optic would not address the encountered challenges of the assessment of medial joint aspects in dogs. While acknowledging the potential benefits of using the 30° optic, we maintain that it is not a viable solution to the issue at hand. Furthermore, it can be assumed that using an angulated optic would obstruct the view of the medial structures, which can only be achieved by straight advancement of the arthroscope due to the cylindrical setup of the joint compartments. It is presumably especially beneficial for inexperienced examiners to use a 0° optic, as it offers an easier orientation during arthroscopy, thus simplifying the learning and application of this diagnostic method.

It should be noted that the arthroscopy conditions in the present study were quite different from a diseased joint. The cadavers used did not show any TMJ disease in advance, which resulted in good visibility of the anatomical structures. The anomalies shown were only visible in individual TMJs and were characterised by changes in the tissue structures within the joint, and impaired vision in some situations (Figure 6).

In cases where TMJ arthroscopy is performed, this condition might not apply. Especially in cases of traumatic injuries, degenerative joint diseases, inflammatory processes, or congenital anomalies, considerable deviations of the joint structure and synovial fluid are to be expected [10,11,25]. Especially in these cases, the examination scheme developed in this study offers the surgeon the possibility of orientation.

Exploration of the joint cavity can be challenging for the surgeon because of the narrow TMJ cavity in animals [21]. Because the initial focus of the study was to evaluate the exploration scheme, we chose to use an arthrotomic approach to the joint. This was achieved by a horizontal incision lateral to the joint. Nevertheless, the joint access was kept as small as possible to only allow penetration of the joint by arthroscope. The trial showed that the selected structures to be examined in dogs could be assessed with high reliability. For this reason, this study can be considered successful.

The resulting surgical wound can be closed using a routine three-layer closure subsequently without drainage as it is supposed to be a clean procedure [25]. Previous studies also suggest that a rapid return of physiologic function can be achieved by performing an arthrotomy of a diseased TMJ, and thus the benefits might outweigh the potential risks [25]. Since minimally invasive arthroscopic joint access offers significant advantages over arthrotomy, such as reduced soft-tissue trauma, faster and better postoperative recovery, and reduced postoperative morbidity, a follow-up study should examine the success of TMJ exploration using arthroscopic joint access [17,33,34,35].

Due to the freezing and thawing process in all specimens prior to the study, it can be assumed that soft tissues have undergone some changes because of this process. Despite these changes, the conditions were assessed uniformly for all cadavers. Several studies in the past have shown that the freezing rate, as well as the thawing rate, do affect the tissues [36,37]. Articular cartilage stiffness, the tension in the tissue, and the strain energy density, decrease. Although TMJ arthroscopy was found feasible in the present study, a clinical trial is required to confirm the efficacy despite histological studies showing no obvious differences between fresh and frozen specimens [37].

Despite this, the results are excellent, so it cannot be assumed that the test series would have achieved adequately better results without the prior freezing process. Nevertheless, the test procedure may behave differently with freshly dead animals or even live animals and should therefore be tested in a subsequent study.

## 5. Conclusions

The investigative protocol developed in this study for arthroscopic examination of the TMJ in dogs proved successful among canines weighing over 20 kg selected for this investigation. In most cadavers, all 15 anatomical structures of both TMJ cavities could be visualised using the needle arthroscope, potentially offering clinicians an advantage in implementing this instrument for diagnosing patients suspected of TMJ disorders. A particular challenge encountered was examining the medial aspect of the joint in dogs which were heavily muscled, as observed in Rhodesian Ridgebacks in this study. Consequently, further research focusing on various breeds to delineate potential interbreed differences is warranted. Likewise, studies involving small dogs and cats should be undertaken to further validate and assess this arthroscopic guidance protocol.

## Figures and Tables

**Figure 1 animals-14-01338-f001:**
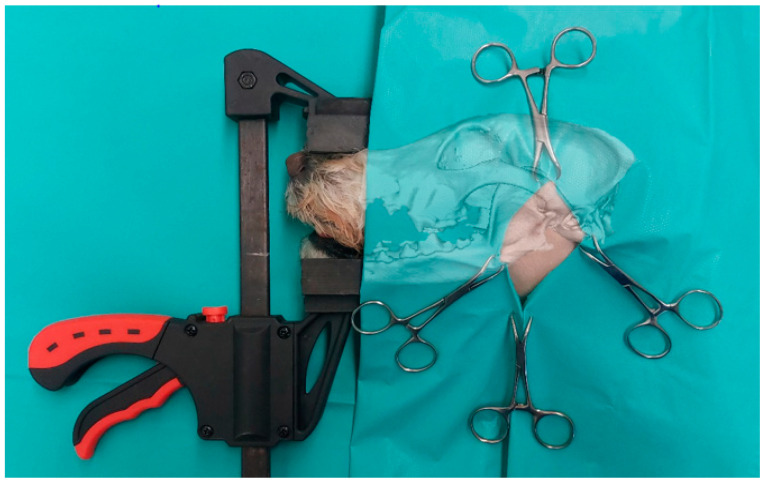
Illustration of the examination setup in a dog positioned in right lateral recumbency. The maxillary and mandibular splints were affixed to the maxilla and mandible of the animal using a quick-grip clamp. To enhance clarity, a semitransparent bone structure was superimposed onto the image.

**Figure 2 animals-14-01338-f002:**
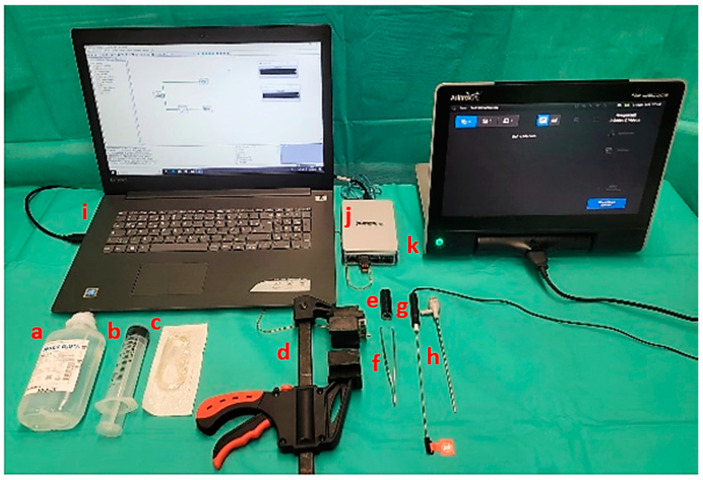
Illustration of the experimental setup and the materials to be used: (a) 0.9% NaCl flushing solution, (b) 60 mL syringe, (c) infusion tubing to facilitate fixation of the syringe with the flushing attachment, (d) quick-grip clamp, incl. inserted upper and lower jaw splint with integrated measuring sensor, (e) tubular element with a diameter of 1 cm, surrounded with insulating tape, (f) surgical forceps, (g) NanoScope Handpiece (diameter 1.9 mm), (h) NanoScope sleeve (diameter 2.4 mm), (i) laptop for continuous measurement of force with integrated program, (j) force transmission device, (k) screen of the needle arthroscope.

**Figure 3 animals-14-01338-f003:**
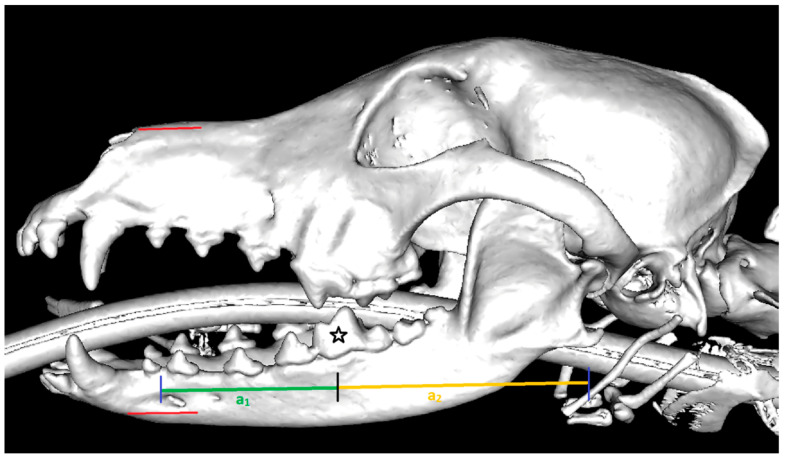
Illustration of the structures to be measured for calculating the force applied to the jaw. The red lines show the position of the upper and lower jaw moulds, including the built-in force sensor, placed on the canine head; the black star shows the first molar tooth on which the tubular element was placed during the examination; the green line indicates the distance a_1_ to be measured (centre of the force sensor to the first molar tooth); the yellow line describes distance a_2_ (first molar to the TMJ); the black line symbolises the transition between a_1_ and a_2_; the purple lines describe the beginning of line a_1_ and the end of distance a_2_ to be measured.

**Figure 4 animals-14-01338-f004:**
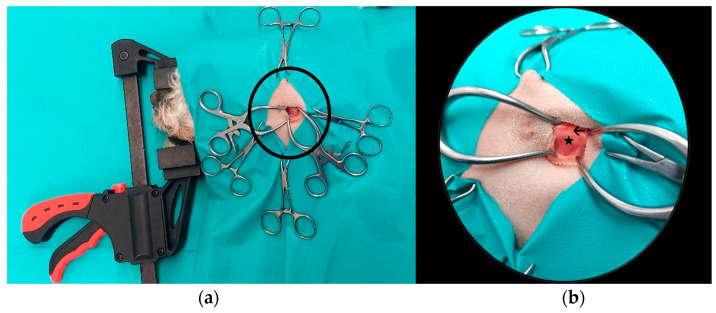
Illustration of a dog in the right lateral position, with a view of the left side of the body with extended access of the left TMJ for simplified visualisation of the anatomical structures, distant view (**a**), close-up (**b**). The caput mandibulae (star) with the overlying TMJ covered by soft tissue (arrow).

**Figure 5 animals-14-01338-f005:**
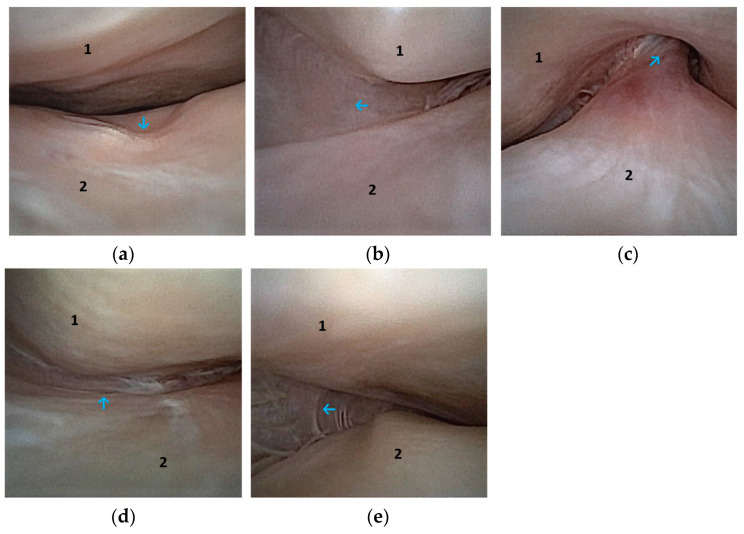
Arthroscopic image of the upper joint (DTJ); the temporal bone (1) is visible dorsally and the articular disc (2) ventrally. The left temporomandibular joint, centre of the joint (blue arrow), (**a**). Cranial aspect of the joint and attachment of the disc visible, left joint (blue arrow), (**b**). Medial attachment of the disc visible, left joint (blue arrow), (**c**). Medial aspect of the joint, left joint (blue arrow), (**d**). Caudal aspect of the joint and insertion of the disc visible, right joint (blue arrow), (**e**).

**Figure 6 animals-14-01338-f006:**
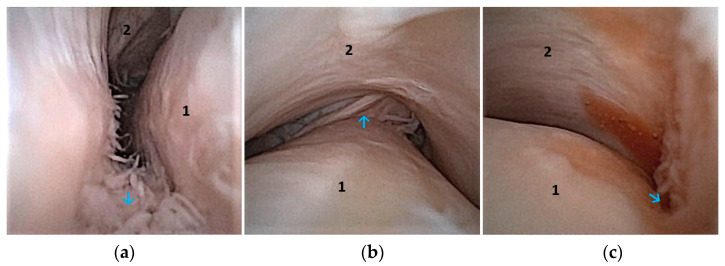
Illustration of the various landmarks examined in the DMJ of a left TMJ during the arthroscopy examination. Caput mandibulae bounds the joint ventrally (1). *Discus articularis* limits the joint dorsally (2). Cranial aspect of the joint (blue arrow), (**a**). Middle aspect of the joint, retro-discal soft tissue, straight back (blue arrow), (**b**). Caudal synovial plicae (blue arrow), (**c**).

**Figure 7 animals-14-01338-f007:**
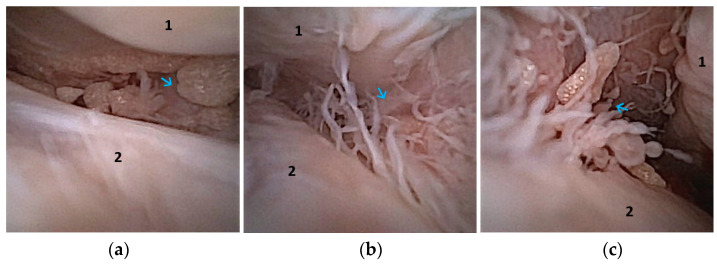
Illustration of anomalies in the DTJ of the left TMJ observed during arthroscopy in three different joints; the temporal bone (1) is visible dorsally and the articular disc (2) ventrally. Tissue changes in the temporomandibular joint are recognisable (blue arrow). Caudal aspect and insertion of the articular disc, visible (**a**,**b**). Cranial aspect of the joint and insertion of the discus are apparent (**c**).

**Table 1 animals-14-01338-t001:** The anatomical structures shown represent the sum of both temporomandibular joints (left and right) for each of the 32 dogs. In total, a maximum of 64 structures were subjected to an applied force of 50–55 newtons. Listed are the 15 different structures to be examined.

Compartment	Anatomical Structure	Visualised Observer 1 (%)	Visualised Observer 2 (%)
Dorsal	Rostral aspect of joint	63 (98%)	63 (98%)
Dorsal	Medial aspect of joint	59 (92%)	55 (86%)
Dorsal	Caudal aspect of joint	64 (100%)	64 (100%)
Dorsal	*Fossa mandibularis* of the *Os temporale*	64 (100%)	64 (100%)
Dorsal	*Discus articularis*	64 (100%)	64 (100%)
Dorsal	Retro-discal tissue	64 (100%)	64 (100%)
Dorsal	Rostral attachment of the *Discus articularis*	62 (97%)	62 (97%)
Dorsal	Medial attachment of the *Discus articularis*	61 (95%)	60 (94%)
Dorsal	Caudal attachment of the *Discus articularis*	62 (97%)	62 (97%)
Ventral	Rostral aspect of joint	62 (97%)	62 (97%)
Ventral	Medial aspect of joint	60 (94%)	60 (94%)
Ventral	Caudal synovial plicae	62 (97%)	62 (97%)
Ventral	Caput mandibulae	64 (100%)	64 (100%)
Ventral	*Discus articularis*	64 (100%)	64 (100%)
Ventral	Retro-discal tissue	64 (100%)	64 (100%)

**Table 2 animals-14-01338-t002:** Percentage of visible anatomical structures during a force development of 50–55 newtons and 100 newtons for observer 1 and observer 2.

Compartment	Force in Newton	Number of Anatomical Structures Visualised	
		Observer 1	Observer 2
Dorsal (n = 576)	50–55	563 (97.74%)	558 (96.88%)
	100	565 (98.09%)	560 (97.22%)
Ventral (n = 384)	50–55	376 (97.92%)	376 (97.92%)
	100	380 (98.96%)	380 (98.96%)

## Data Availability

The data presented in this study are available upon request from the corresponding authors.

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
