# Peer review of "Development and Validation of an Examination Protocol for Arthroscopic Evaluation of the Temporomandibular Joint in Dogs"

_animals, 2024, doi:10.3390/ani14091338_

Round 1
Reviewer 1 Report
Comments and Suggestions for Authors
Comments and Suggestions for Authors:
Line 2-3: Replace “Development and validation of an examination procedure for the arthroscopic temporomandibular joint examination in dogs” for “Development and validation of an examination protocol for arthroscopic evaluation of the temporomandibular joint in dogs”.
Line 36: Add the keyword "dog", and if only 4 words are allowed remove “Veterinary surgery”.
Lines 40-42: The references on the use of computed tomography (CT) in the evaluation of TMJ in small animals seem insufficient. In the bibliography, there are numerous references on the use of this technique in TMJ evaluation, many of them more recent than the studies referenced in the paper. The authors should explain the advantages and disadvantages of computed tomography in TMJ evaluation. CT is less invasive than arthroscopy. Arthroscopy allows for the evaluation of articular cartilage, the disc, and the soft tissues of the joint. Based on this, they should refine the hypothesis of their study and discuss the advantages of arthroscopy over CT in the discussion.
Lines 44-46: Using a reference on equine anatomy to explain the anatomy of the temporomandibular joint (TMJ) in dogs is not appropriate. Consider changing and/or adding citations instead of reference number 7.
Lines 46-48: To describe the anatomy of a region, it would be appropriate to use anatomical references instead of citations from clinical papers. It is possible that those papers use more appropriate citations. Consider replacing them.
Lines 67-68: References 16 and 17 refer to experimental studies conducted on dogs. This aspect should be clarified.
Line 105. Include the references for the decision to apply a distraction force of 50-55 newtons. It is explained in the discussion (lines 301-310), but it would be clearer that it is not a randomly chosen force.
Lines 109-111: To facilitate the reproduction of the study, it would be better to place a detailed image of the placement of the splints on the rostral portion of the mandible and maxilla as Figure 1. The current Figure 1 would become Figure 2, and it would explain the calculation of the force applied to the mandible.
Lines 138-148: Is this paragraph describing an arthrotomy of the TMJ followed by an assessment of the joint with an arthroscope? Is it indeed so? In the evaluation of the TMJ, is the lateral aspect assessed by arthrotomy and the medial aspect by pure arthroscopy?. In lines 338-339, it is stated, "the access to the joint was kept as small as possible to allow only the penetration of the joint by the arthroscope”. Add a reference to this aspect in the Materials and Methods section, although it will be further elaborated in the discussion.
Line 149: The two images in Figure 2 (2a and 2b) have very poor quality. They should be enlarged and improved in quality. They should be divided into Figures 3 and 4. Currently, there are very few differences between Figure 2b and 3a, except for the approach. I would recommend eliminating Figure 3a.
Lines 187-188: Subsequently, in the results section, there is mention of 2 observers. Clarify whether the examiner and the assistant were observer 1 and observer 2. Replace “All arthroscopies were performed by the same examiner and assistant (observer) to ensure consistency of results” for “All arthroscopies were performed by the same examiner (observer 1) and assistant (observer 2) to ensure consistency of results.”
At this point, the question arises as to whether both conducted the evaluation jointly, and how it is determined that the assistant was not influenced by the examiner's assessment. It could have been more effective to record the arthroscopies and allow other observers to evaluate them. This would have provided an opportunity for a more detailed review and independent assessment of the findings, which could have improved the validity and reliability of the results. Additionally, it would have allowed for a more thorough review of the procedures and provided a visual reference for future research and training in canine TMJ arthroscopy techniques.
Line 211. At this point, the list of evaluated anatomical structures should be included (independent of the table 1).
Table 1: Replace “medial attachment of the Discus articularis” for “Medial attachment of the Discus articularis” and “caudal attachment of the Discus articularis” for “Caudal attachment of the Discus Articularis”
Table 2: Given that the differences are minimal, two decimal places should be added to all percentages for Observer 1 and Observer 2.
Lines 288-290. It doesn't make much sense to recommend arthroscopy for the diagnosis of certain TMJ pathologies and then cite a study that uses CT for diagnosis (reference 3). It would be interesting to look for other pathologies where CT is not as effective.
Lines 330-338. Personally, I would move this paragraph to an earlier section to better explain why applying distraction force is necessary.
Lines 412-417. In the conclusions, some specific aspects of the performed arthroscopic technique should be included, not just the indications.
Lines 412-413. The statement " The examination scheme developed in the present study for arthroscopic TMJ examination in dogs was successful." could be misleading since the average weight of the animals in this study is not clarified. Although there is a reference to small dogs and cats in the last sentence of the conclusion, that data is important.
Line 415: Remove “(Table 1”) from the sentence.
Author Response
Dear Reviewer 1,
Thank you for giving us the opportunity to submit a revised draft of the manuscript. We appreciate the time and effort you have dedicated to providing your valuable feedback on the manuscript. Enclosed herein is a point-by-point response to your comments. The line numbers referenced pertain to the newly corrected and uploaded manuscript.
Kind regards,
Ina Quadflieg
Point-by-point response to Comments and Suggestions:
Comment 1:
Line 2-3: Replace “Development and validation of an examination procedure for the arthroscopic temporomandibular joint examination in dogs” for “Development and validation of an examination protocol for arthroscopic evaluation of the temporomandibular joint in dogs”.
Response 1:
Thank you for your valuable advice, we have adjusted the headline as requested (Lines 2-3).
Comment 2:
Line 36: Add the keyword "dog", and if only 4 words are allowed remove “Veterinary surgery”.
Response 2:
Thank you for pointing this out, we have added the suggested keyword (Lines 38).
Comment 3:
Lines 40-42: The references on the use of computed tomography (CT) in the evaluation of TMJ in small animals seem insufficient. In the bibliography, there are numerous references on the use of this technique in TMJ evaluation, many of them more recent than the studies referenced in the paper. The authors should explain the advantages and disadvantages of computed tomography in TMJ evaluation. CT is less invasive than arthroscopy. Arthroscopy allows for the evaluation of articular cartilage, the disc, and the soft tissues of the joint. Based on this, they should refine the hypothesis of their study and discuss the advantages of arthroscopy over CT in the discussion.
Response 3:
Thank you for your comment. We have identified additional resources on temporomandibular joint diagnostics using CT and presented the advantages and disadvantages of the various diagnostic tools. The introductory and discussion sections have been modified as you suggested (Lines 62-82, 346-353)
Comment 4:
Lines 44-46: Using a reference on equine anatomy to explain the anatomy of the temporomandibular joint (TMJ) in dogs is not appropriate. Consider changing and/or adding citations instead of reference number 7.
Response 4:
Thank you very much for this insightful reference. We have added another reference to this section that discusses the anatomy of the temporomandibular joint (Line 48)
Comment 5:
Lines 46-48: To describe the anatomy of a region, it would be appropriate to use anatomical references instead of citations from clinical papers. It is possible that those papers use more appropriate citations. Consider replacing them.
Response 5:
Thank you for your comment. We have added another anatomical reference here to explain the anatomy of the temporomandibular joint (Line 50)
Comment 6:
Lines 67-68: References 16 and 17 refer to experimental studies conducted on dogs. This aspect should be clarified.
Response 6:
Thank you for this remark. We have added this aspect to the manuscript and hope that this section is now easier for the reader to understand (Lines 83-85)
Comment 7:
Line 105. Include the references for the decision to apply a distraction force of 50-55 newtons. It is explained in the discussion (lines 301-310), but it would be clearer that it is not a randomly chosen force.
Response 7:
Thank you for this note. We have added an explanation of the selected force setting in the material and method section (Lines 122-124)
Comment 8:
Lines 109-111: To facilitate the reproduction of the study, it would be better to place a detailed image of the placement of the splints on the rostral portion of the mandible and maxilla as Figure 1. The current Figure 1 would become Figure 2, and it would explain the calculation of the force applied to the mandible.
Response 8:
Thank you for this comment. The changes you mentioned have been made and the images have been rearranged as you suggested (Figure 1/2/3)
Comment 9:
Lines 138-148: Is this paragraph describing an arthrotomy of the TMJ followed by an assessment of the joint with an arthroscope? Is it indeed so? In the evaluation of the TMJ, is the lateral aspect assessed by arthrotomy and the medial aspect by pure arthroscopy?. In lines 338-339, it is stated, "the access to the joint was kept as small as possible to allow only the penetration of the joint by the arthroscope”. Add a reference to this aspect in the Materials and Methods section, although it will be further elaborated in the discussion.
Response 9:
Thank you for this request. As you have already correctly mentioned, the initial arthroscopic approach was only used to make it easier to find and enter the selected joint cavity. For this reason, the examination access was kept as small as possible and only a small stab incision was made in both joint cavities to facilitate penetration into the joint. Therefore, all anatomical structures of the joint (both medial and lateral structures) were examined arthroscopically. To clarify the intention of the procedure to the reader, a sentence has been added to the materials and methods section (Lines 174-177)
Comment 10:
Line 149: The two images in Figure 2 (2a and 2b) have very poor quality. They should be enlarged and improved in quality. They should be divided into Figures 3 and 4. Currently, there are very few differences between Figure 2b and 3a, except for the approach. I would recommend eliminating Figure 3a.
Response 10:
Thank you kindly for your valuable input. We have adjusted the image size and quality of the original image 2a and hope that this image has now been improved to your approval. We have also decided to adjust the image arrangement as noted in your previous comment. Thus, we have deleted image 2b from the manuscript and images 3a and 3b have been retained, as the clarification of the close-up of image 3b from the original image 3a was important for the comprehensibility of the study (Figure 1,2,3)
Comment 11:
Lines 187-188: Subsequently, in the results section, there is mention of 2 observers. Clarify whether the examiner and the assistant were observer 1 and observer 2. Replace “All arthroscopies were performed by the same examiner and assistant (observer) to ensure consistency of results” for “All arthroscopies were performed by the same examiner (observer 1) and assistant (observer 2) to ensure consistency of results.”
Response 11:
Thank you very much for this comment. You are absolutely right that your proposed sentence is much easier to understand, this has been addressed as suggested (Lines 230-231)
Comment 12:
At this point, the question arises as to whether both conducted the evaluation jointly, and how it is determined that the assistant was not influenced by the examiner's assessment. It could have been more effective to record the arthroscopies and allow other observers to evaluate them. This would have provided an opportunity for a more detailed review and independent assessment of the findings, which could have improved the validity and reliability of the results. Additionally, it would have allowed for a more thorough review of the procedures and provided a visual reference for future research and training in canine TMJ arthroscopy techniques.
Response 12:
Thank you for the request and the information. The results were analysed by both investigators independently of each other. The examiners proceeded as follows: First, the arthroscopy of the respective animal was performed bilaterally and both examiners recorded their results independently. During the individual examinations, it was therefore very important that there was no communication between the two examiners in order to avoid any influence. After each examination, it was determined whether the examiners could assess the selected structures, in order to potentially increase the force applied to the jaw if necessary.
Thank you for the constructive suggestion to bring in a separate observer to evaluate the video material. We also considered this, but decided against it, as the evaluation of pure video material represents an increased difficulty due to the possibility of orientation and also the transmitted image quality and thus the comparability to the direct findings during the results could be falsified (Lines 234-241).
Comment 13:
Line 211. At this point, the list of evaluated anatomical structures should be included (independent of the table 1).
Response 13:
Thank you for this indication. A text passage has been added from Lines 261-267, listing the chosen anatomical structures.
Comment 14:
Table 1: Replace “medial attachment of the Discus articularis” for “Medial attachment of the Discus articularis” and “caudal attachment of the Discus articularis” for “Caudal attachment of the Discus Articularis”
Response 14:
Thank you very much for correcting the formal mistake in Table 1, your comment has been amended (Table 1).
Comment 15:
Table 2: Given that the differences are minimal, two decimal places should be added to all percentages for Observer 1 and Observer 2.
Response 15:
Thank you for this annotation, we have added 2 decimal numbers to the table to make the difference between the two observers easier to understand (Table 2).
Comment 16:
Lines 288-290. It doesn't make much sense to recommend arthroscopy for the diagnosis of certain TMJ pathologies and then cite a study that uses CT for diagnosis (reference 3). It would be interesting to look for other pathologies where CT is not as effective.
Response 16:
Thank you for your remark. The text passage has been amended to include a reference to the benefits of arthroscopic examination (Lines 346-353).
Comment 17:
Lines 330-338. Personally, I would move this paragraph to an earlier section to better explain why applying distraction force is necessary.
Response 17:
Thank you for your comment, which we appreciate. However, we did not find a better place in the manuscript without interrupting the flow.
Comment 18:
Lines 412-417. In the conclusions, some specific aspects of the performed arthroscopic technique should be included, not just the indications.
Response 18:
Thank you for this addition. We have revised the summary and added specific aspects of the arthroscopic examination (Lines 486-495).
Comment 19:
Lines 412-413. The statement " The examination scheme developed in the present study for arthroscopic TMJ examination in dogs was successful." could be misleading since the average weight of the animals in this study is not clarified. Although there is a reference to small dogs and cats in the last sentence of the conclusion, that data is important.
Response 19:
Thank you for this reference. Subsequently, it was clarified in the summary that this study scheme has been successful for animals >20kg and reference was made to further studies on different breeds, smaller dogs and also cats (Lines 486-495).
Comment 20:
Line 415: Remove “(Table 1”) from the sentence.
Response 20:
Thank you for pointing this out, “Table 1” has been removed from the sentence (Lines 486-495)
Reviewer 2 Report
Comments and Suggestions for Authors
Dear Authors,
I included in the manuscript review attached the my impressions and suggestions to improve your paper including the special attention to the latin terms which ought to be in italic.
I also wanted to clarify if there is a difference between the spayed male and female animals concerning the joint accessibility and anatomical structure analysis.
Best regards,

Comments on the Quality of English LanguageThe English must be reviewed for some small issues. Overall quality is fine.
Author Response
Dear Reviewer 2,
Thank you for giving us the opportunity to submit a revised draft of the manuscript. We appreciate the time and effort you have dedicated to providing your valuable feedback on the manuscript. Enclosed herein is a point-by-point response to your comments. The line numbers referenced pertain to the newly corrected and uploaded manuscript.
Kind regards,
Ina Quadflieg
Point-by-point response to Comments and Suggestions:
Comment 1:
I included in the manuscript review attached the my impressions and suggestions to improve your paper including the special attention to the latin terms which ought to be in italic.
Response 1:
Thank you for your valuable comments, we have adjusted the Latin terms in the manuscript and will respond to your further comments below.
Comment 2:
I also wanted to clarify if there is a difference between the spayed male and female animals concerning the joint accessibility and anatomical structure analysis.
Response 2:
Thank you for this request. There were no differences between neutered males and spayed females in the studies. This is an interesting point that should be addressed in a larger-scale follow-up study. However, this would require determining the exact castration age of the animals and comparing them to intact animals. This was beyond the scope of the current study.
Comment 3:
Is there any anatomical difference between the listed breeds that might interfere in the study concerning the results and statistical analysis?
Response 3:
Thank you for this comment. Some studies have shown that there can be different characteristics of the temporomandibular joint within the skull morphologies and skull sizes. However, it has been found that small brachycephalic dogs are particularly affected and can suffer from incongruent temporomandibular joints and a poorly developed or absent retroarticular process. This can lead to laxity of the temporomandibular joint, as the stability of the temporomandibular joint is reduced (1, 2) It can be assumed that this facilitates access to the joint, as the existing TMJ structures are more easily accessible based on this reshaping and increased laxity. Therefore, a follow-up arthroscopic study with small dogs would be useful to investigate this hypothesis. The dogs included in this study all had well-integrated temporomandibular joints, so arthroscopic examination of these animals was probably not affected or influenced by breed. We therefore assume that the results did not influence the analysis and statistics. Future studies are necessary to address this in more detail. Based on your comments we have expanded the discussion section (Lines 409-419)
Comment 4:
For a functional study which are the signals and symptoms to consider in the canines for a arthroscopy examination?
Response 4:
Thank you for this comment. The symptoms and signals of the animals can be diverse, ranging from pain to movement restrictions to joint dysfunction (Lines 55-57)
Comment 5:
By performing the arthroscopy does the therapeutical approach change?
Response 5:
Thank you for this request. The therapeutic approach should not necessarily change as a result of TMJ arthroscopy, but should offer an additional therapeutic option. It must be decided on an individual basis which approach is best for the animal in question. It is not wrong to initially attempt conservative treatment in affected animals, as this is significantly less invasive. In cases of recurrent or persistent temporomandibular joint disorders, consideration should be given to arthroscopy, and this approach can also be chosen directly in cases of septic arthritis, as direct assessment of the joint and simultaneous irrigation have proven to be beneficial (3). In addition to arthroscopy, medical therapy with analgesics and, depending on the cause and severity of the disease, antibiotic therapy must be administered.
- Curth S, Fischer MS, Kupczik K. Can skull form predict the shape of the temporomandibular joint? A study using geometric morphometrics on the skulls of wolves and domestic dogs. Annals of Anatomy-Anatomischer Anzeiger. 2017;214:53-62.
- Villamizar-Martinez LA, Villegas CM, Gioso MA, Reiter AM, Patricio GC, Pinto AC. Morphologic and morphometric description of the temporomandibular joint in the domestic dog using computed tomography. Journal of veterinary dentistry. 2016;33(2):75-82.
- Arzi B, Vapniarsky N, Fulton A, Verstraete F. Management of Septic Arthritis of the Temporomandibular Joint in Dogs. Frontiers in Veterinary Science. 2021;8:1-9.
Round 2
Reviewer 1 Report
Comments and Suggestions for Authors
The authors have made the clarifications and changes requested in the first review, improving the clarity and comprehension of the text.
I consider the work is ready for publication.